# Dopamine and acetylcholine have distinct roles in delay- and effort-based decision-making in humans

**Mani Erfanian Abdoust**[1]*, **Monja Isabel Froböse**[1], **Alfons Schnitzler**[2], **Elisabeth Schreivogel**[2], **Gerhard Jocham**[1]

1 Biological Psychology of Decision Making, Institute of Experimental Psychology, Heinrich Heine University Düsseldorf, Düsseldorf, Germany, 2 Institute of Clinical Neuroscience and Medical Psychology, Medical Faculty, Heinrich Heine University Düsseldorf, Germany

* erfanian@hhu.de

**Data Availability Statement:** Anonymized and preprocessed datasets, along with the regression and computational modelling analysis codes, are available at https://osf.io/gx76t/.

## Abstract

In everyday life, we encounter situations that require tradeoffs between potential rewards and associated costs, such as time and (physical) effort. The literature indicates a prominent role for dopamine in discounting of both delay and effort, with mixed findings for delay discounting in humans. Moreover, the reciprocal antagonistic interaction between dopaminergic and cholinergic transmission in the striatum suggests a potential opponent role of acetylcholine in these processes. We found opposing effects of dopamine D2 (haloperidol) and acetylcholine M1 receptor (biperiden) antagonism on specific components of effort-based decision-making in healthy humans: haloperidol decreased, whereas biperiden increased the willingness to exert physical effort. In contrast, delay discounting was reduced under haloperidol, but not affected by biperiden. Together, our data suggest that dopamine, acting at D2 receptors, modulates both effort and delay discounting, while acetylcholine, acting at M1 receptors, appears to exert a more specific influence on effort discounting only.

## Introduction

Consider the following scenarios: Would you prefer to order an average pizza that can be delivered within minutes or rather wait an extra hour for your most favourite pizza? Similarly, would you visit a highly rated pizza place that requires climbing up a steep hill or opting for a conveniently accessible pizza place right in front of the next bus stop? In scenarios like this, decision-making involves the balancing of potential rewards against distinct costs required to obtain them. The tendency to devalue rewards as a function of effort or delay costs are commonly described as effort and delay discounting, respectively [1,2].

Striatal dopamine has been suggested to play a central role in both effort and delay discounting. For effort-based decision-making, studies in both humans and rodents indicate that diminished dopamine transmission makes individuals less willing to exert physical effort in exchange for larger rewards [3–9]. Pharmacological studies consistently demonstrate that agents that enhance dopaminergic transmission in humans increase reward sensitivity and

**Funding:** This work was supported by a grant (DFG JO-787/5-1 to GJ) from the Deutsche Forschungsgemeinschaft (DFG; https://www.dfg. de/). The funders had no role in study design, data collection and analysis, decision to publish, or preparation of the manuscript.

**Competing interests:** The authors have declared that no competing interests exist.

**Abbreviations:** AES, Apathy Evaluation Scale; BDI, Beck's Depression Inventory; BIP, biperiden; BIS, Barratt Impulsiveness Scale; GLMM, generalized linear mixed model; HAL, haloperidol; HDI, highest density interval; LOOIC, leave-one-out information criterion; MVC, maximum voluntary contraction; PLC, placebo; SD, standard deviation; SV, subjective value.

decrease effort sensitivity, while studies directly investigating the effects of selective D2-receptor antagonism in healthy humans are rare [10–12]. For delay discounting, the existing literature is somewhat more mixed. Some studies suggest that increased dopamine transmission decreases delay discounting [13–15], whereas others have found either no effect [16,17] or an increase in delay discounting [18]. Notably, more recent evidence in humans indicates decreased delay discounting after blockade of D2 receptors [19,20]. While dopamine has received much attention in cost-benefit decision-making, other neuromodulators may also play an important role [21–23]. One modulator that has not received much attention in this regard is acetylcholine, which is surprising given the literature on reciprocal antagonistic interactions between acetylcholine and dopamine signalling in the striatum [24–28]. This suggests that pharmacological blockade of M1 receptors will have effects on cost-benefit decision-making that are opposite to those of blocking dopamine D2 receptors. In line with this, animal studies indicate that muscarinic agonists induce behavioural changes in effort-based choices that are similar to those produced by dopamine antagonists [23,29], underscoring the potential interplay between dopamine and acetylcholine in modulating cost-benefit decision-making. Furthermore, research in animal models points to a role for acetylcholine in modulating delay-based choices through both muscarinic and nicotinic receptors, albeit with inconsistent findings [30–33]. Importantly, studies directly investigating these effects in human decision-making are lacking.

To our knowledge, there has been no study so far that tested the impact of dopaminergic and cholinergic manipulations on both aspects of cost-benefit decision-making in 1 single experiment. To fill this gap, we investigated the effects of 2 drugs, haloperidol and biperiden, that selectively block either dopamine D2-like or muscarinic M1 acetylcholine receptors in human participants performing 2 decision-making tasks involving effort- and delay-based decisions. The goal of our study was 3-fold. First, we aimed to conceptually replicate the finding that dopamine D2 antagonists increase discounting of physical effort. Second, we aimed to assess the contribution of acetylcholine to cost-benefit decision-making and conceptually contrast it with the effects of dopamine. In doing so, we investigated whether any effects of muscarinic or dopaminergic receptor antagonism would modulate the computation of both time and effort costs or have a more specific effect on one cost dimension. Third, we sought to contribute new evidence to the thus far conflicting literature on the role of dopamine in delay-based decision-making using a large sample size and a within-subjects design. Based on previous findings, we hypothesised that the administration of haloperidol will (1) increase effort discounting; and (2) reduce delay discounting. In contrast, we expected opposite effects of biperiden, in particular (3) a decrease in effort discounting; and (4) an increase in delay discounting. In brief, we found opposing effects of haloperidol and biperiden only on specific components of effort-based choices. Specifically, haloperidol reduced the willingness to invest physical effort, whereas biperiden increased it. Results for delay discounting were less consistent. While haloperidol decreased delay discounting, there was no credible modulation by biperiden.

## Results

The 62 healthy participants performed 2 cost-benefit decision-making tasks aimed to quantify the extent to which the subjective value (SV) of a monetary reward is discounted as a function of either effort or delay costs. Participants completed both tasks during 3 sessions under the influence of either the D2 receptor antagonist haloperidol (2 mg), the M1 acetylcholine receptor antagonist biperiden (4 mg), or a placebo in a within-subjects design (Fig 1A). The effort-based decision-making task (effort discounting task, Fig 1B) involved choices between options

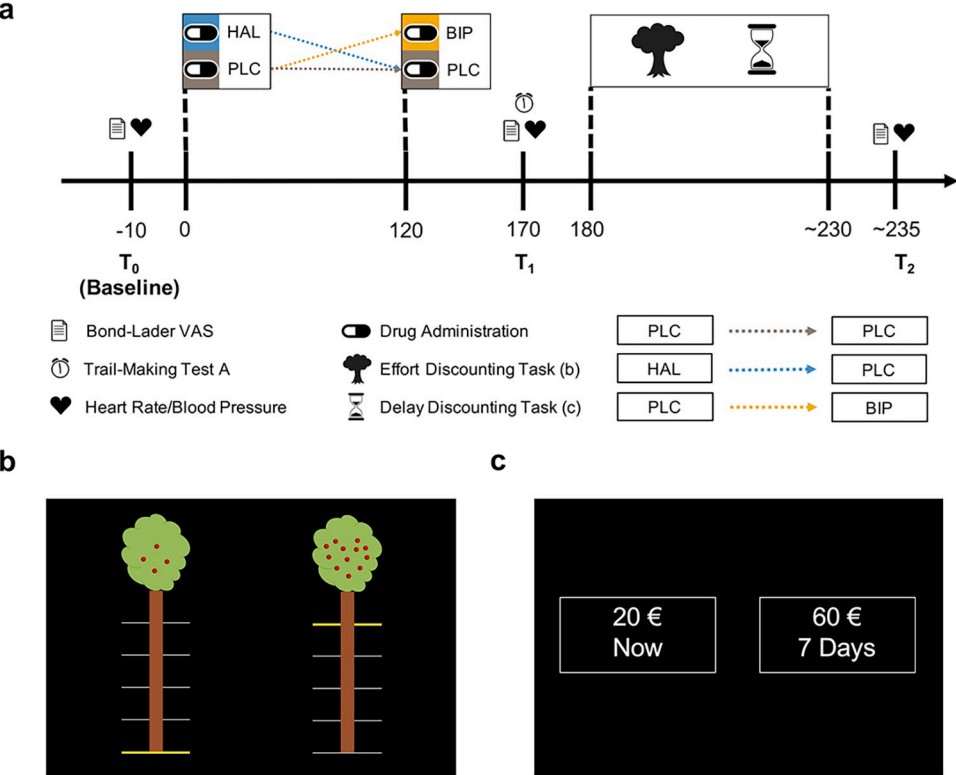

**Fig 1. Study procedure and experimental tasks.** (**a**) The experimental sessions were standardised across all sessions except for the drug treatment, which was counterbalanced. To account for the different times of haloperidol and biperiden to reach peak plasma levels, a dummy drug application was introduced, ensuring that the maximum concentration of both drugs was aligned to task execution. Approximately 180 min after the first capsule and approximately 60 min after the second capsule, participants engaged in the effort discounting task (b), followed by the delay discounting task (c). Physiological measures (including heart rate and blood pressure) and mood ratings (using the Bond and Lader Visual Analogue Scales) were collected at 3 distinct time points. Additionally, participants completed the trail-making test part A before task execution (see Materials and methods for more details). In both tasks, participants were presented with 2 alternative options, each providing information about a monetary reward in return for specific costs. (**b**) Effort discounting task. One option required less effort (indicated by the horizontal yellow line) and provided a smaller reward (indicated by the number of apples, low-cost option), while the other option required more effort and provided a higher reward (high-cost option). Participants then chose 1 option and exerted the required effort (adjusted to the MVC) for at least 1 s. (**c**) Delay discounting task. Similarly, participants were presented with 2 offers: a smaller but immediately available reward (low-cost option) or a larger reward available after the delay indicated (high-cost option). BIP, biperiden; HAL, haloperidol; MVC, maximum voluntary contraction; PLC, placebo.

varying in reward magnitude and effort requirement (using handgrip force), while the delay-based decision-making task (delay discounting task, Fig 1C) involved choices between one option varying in reward magnitude and delay versus a fixed one. Thus, both tasks required participants to choose between a high-reward/high-cost (high-cost option) and a low-reward/low-cost (low-cost option) alternative.

## Drug effects on choice behaviour

Participants consistently exhibited a preference for the high-cost option over the low-cost alternative in both tasks (effort discounting task: 78.50% ± 1.03; delay discounting task: 71.31% ± 1.12; Fig 2A). For each of the 2 tasks, we used a logistic Bayesian generalized linear mixed model (GLMM) to investigate the impact of drug manipulation, changes in task parameters, and the interaction between both, on choosing the high-cost option. Specifically, to

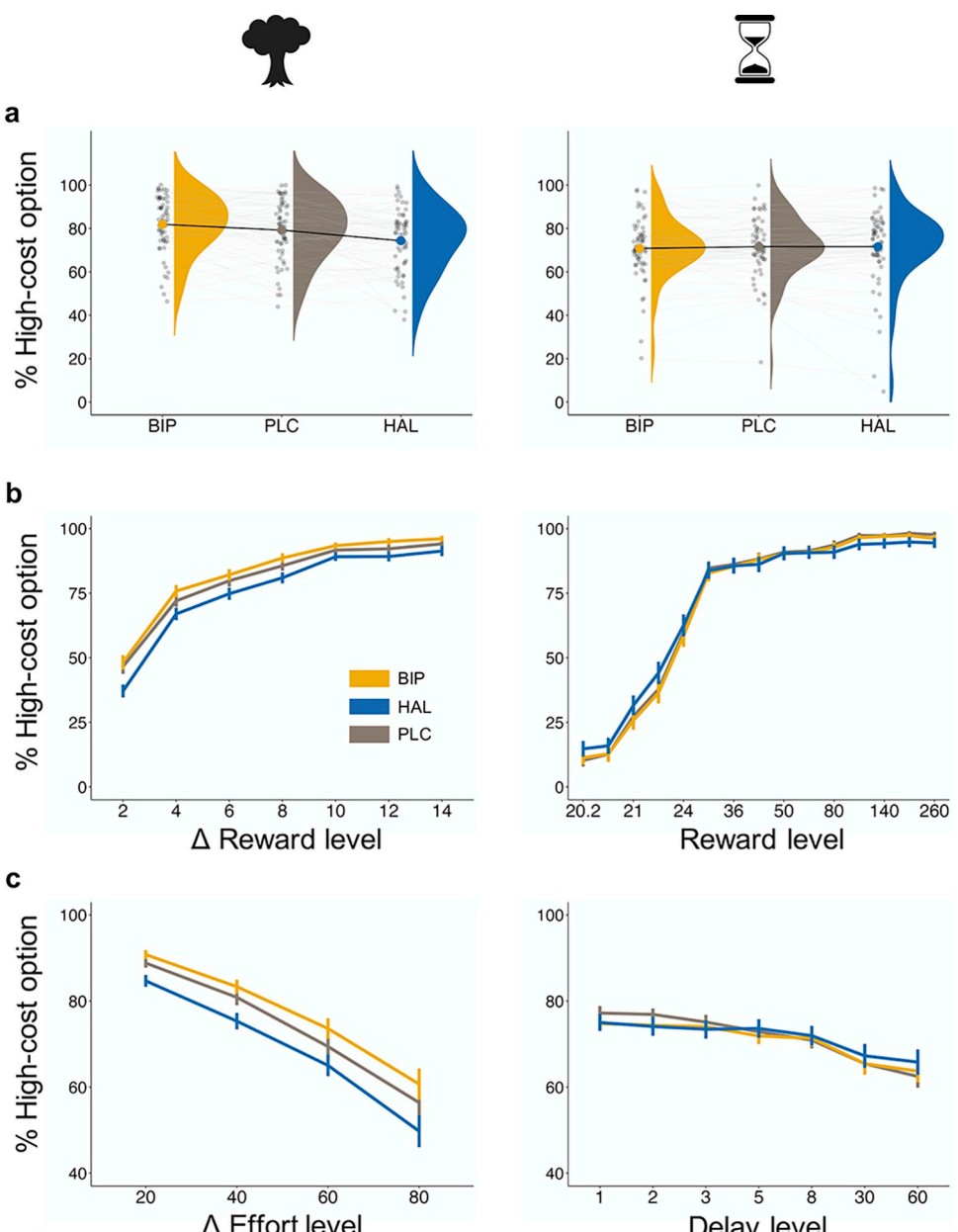

**Fig 2. Behavioural performance in the effort (left panel) and delay (right panel) discounting task.** (**a**) The overall proportions of high-cost choices were modulated by drug administration in the effort (left), but not in the delay discounting task (right). Biperiden increased and haloperidol decreased the willingness to invest physical effort in return for reward. (**b**) The probability of choosing the high-cost option increased as a function of increasing reward magnitude for both the effort and delay discounting tasks. In the effort discounting task (left), biperiden increased the impact of the reward level on choice. (**c**) Similarly, the tendency to choose the high-cost option decreased as a function of increasing levels of effort and delay. This effect is reduced by haloperidol in the delay discounting task (right). Values in (**a**) show group-level (single-subject) means represented by bold (light) dots. Values in (**b**) and (**c**) display averaged group-level means per reward and cost level, with error bars representing the standard error of the mean. In (**b**), reward levels are presented as the difference in magnitude between the high- and low-cost option in the effort discontinuing task and as the absolute reward value of the high-cost option in the delay discounting task. Likewise in (**c**), the effort level represents the difference between the proportions of the individually calibrated MVC of the high- versus low-cost option, while the delay levels indicate the delay of the high-cost option. The data underlying the effort discounting task (left panel) can be found in S1 Data, and the data underlying the delay discounting task (right panel) can be found in S2 Data. MVC, maximum voluntary contraction.

regress participants' choices, we included the following fixed-effect predictors: drug condition, reward magnitude, cost level (i.e., effort or delay), and all possible interactions. In the effort discounting task, we used the difference terms (high-cost option minus low-cost option) for both reward and effort. For the delay discounting task, we used the absolute reward and delay levels of the varying high-cost option. To account for within-subject variability, we included random intercepts for each subject along with random slopes for all fixed-effect predictors (see Materials and methods). We first confirmed that participants effectively discounted rewards based on costs and thereby adhered to the task requirements: As expected, in both discounting tasks higher reward magnitudes increased, while higher cost levels (effort or delay) decreased participants' likelihood to select the high-cost option (reward effect on effort discounting: $HDI_{Mean} = 3.44$, $HDI_{95\%} = [2.96; 3.95]$; reward effect on delay discounting: $HDI_{Mean} = 55.64$, $HDI_{95\%} = [41.76; 69.38]$, effort effect on effort discounting: $HDI_{Mean} = -1.64$, $HDI_{95\%} = [-1.87; -1.40]$; delay effect on delay discounting: $HDI_{Mean} = -2.29$, $HDI_{95\%} = [-3.30; -1.25]$, Fig 2B and 2C).

Notably, and in line with our predictions, we found that the 2 drugs had opposite effects on the tendency to choose high-effort options in the effort discounting task relative to placebo. Haloperidol reduced the willingness to invest higher effort for greater reward, while biperiden increased this willingness (placebo: 79.17% ± 1.76; haloperidol: 74.34% ± 1.85; biperiden: 81.99% ± 1.64; Fig 2A (left panel)). These effects were confirmed by credible main effects of both haloperidol ($HDI_{Mean} = -0.532$, $HDI_{95\%} = [-0.943; -0.136]$) and biperiden ($HDI_{Mean} = 0.620$, $HDI_{95\%} = [0.230; 1.048]$; Fig 3A) on choosing the high-cost option relative to placebo. Moreover, this analysis revealed that biperiden increased reward sensitivity, as evidenced by a credible interaction effect between biperiden and reward ($HDI_{Mean} = 0.802$, $HDI_{95\%} = [0.256; 1.409]$; Fig 3B). See S1 Table for the full results.

In contrast, in the delay discounting task, both drugs had no effect on the average rate of choosing the high-cost option (placebo: 71.59% ± 2.25; haloperidol: 71.55% ± 1.73; biperiden: 70.79% ± 1.82; Fig 2A (right panel)), as indicated by the absence of credible main effects for haloperidol ($HDI_{Mean} = -0.046$, $HDI_{95\%} = [-2.648; 1.884]$) and biperiden ($HDI_{Mean} = -1.315$, $HDI_{95\%} = [-4.147; 0.655]$; Fig 3D). Importantly, however, the analysis revealed a reduced sensitivity to delays under haloperidol, evidenced by a credible interaction effect between haloperidol and delay ($HDI_{Mean} = 1.332$, $HDI_{95\%} = [0.328; 2.440]$; Fig 3F), while the other drug interactions did not reach credibility (Fig 3 and S2 Table). This indicates that diminished dopaminergic activity attenuates the impact of time costs on decision-making.

Having established distinct task-specific drug effects on participants' choice behaviour, we next investigated potential confounding order and fatigue effects using separate GLMMs for effort- and delay-based decision-making. These models mirrored the previously described GLMMs but included additional predictors to test for fatigue and session effects and their modulation by drugs.

To test for possible fatigue effects, we added trial number as well as the two-way interaction with drug as additional predictors. This analysis revealed a credible main effect of trial number for both the effort and delay discounting task (trial number effect on effort discounting: $HDI_{Mean} = -0.60$, $HDI_{95\%} = [-0.75; -0.46]$; trial number effect on delay discounting: $HDI_{Mean} = -0.20$, $HDI_{95\%} = [-0.33; -0.08]$), suggesting that the tendency to choose high-cost options, irrespective of cost type, decreased over the course of the experiment. Notably, we found a credible interaction effect between trial number and haloperidol in the effort discounting task ($HDI_{Mean} = -0.43$, $HDI_{95\%} = [-0.63; -0.23]$), indicating that the fatigue effect was more pronounced under haloperidol compared to placebo. Importantly, even after accounting for these fatigue-related effects, the main effect of haloperidol on effort discounting remained credible ($HDI_{Mean} = -0.48$, $HDI_{95\%} = [-0.90; -0.05]$) (S12 and S13 Tables).

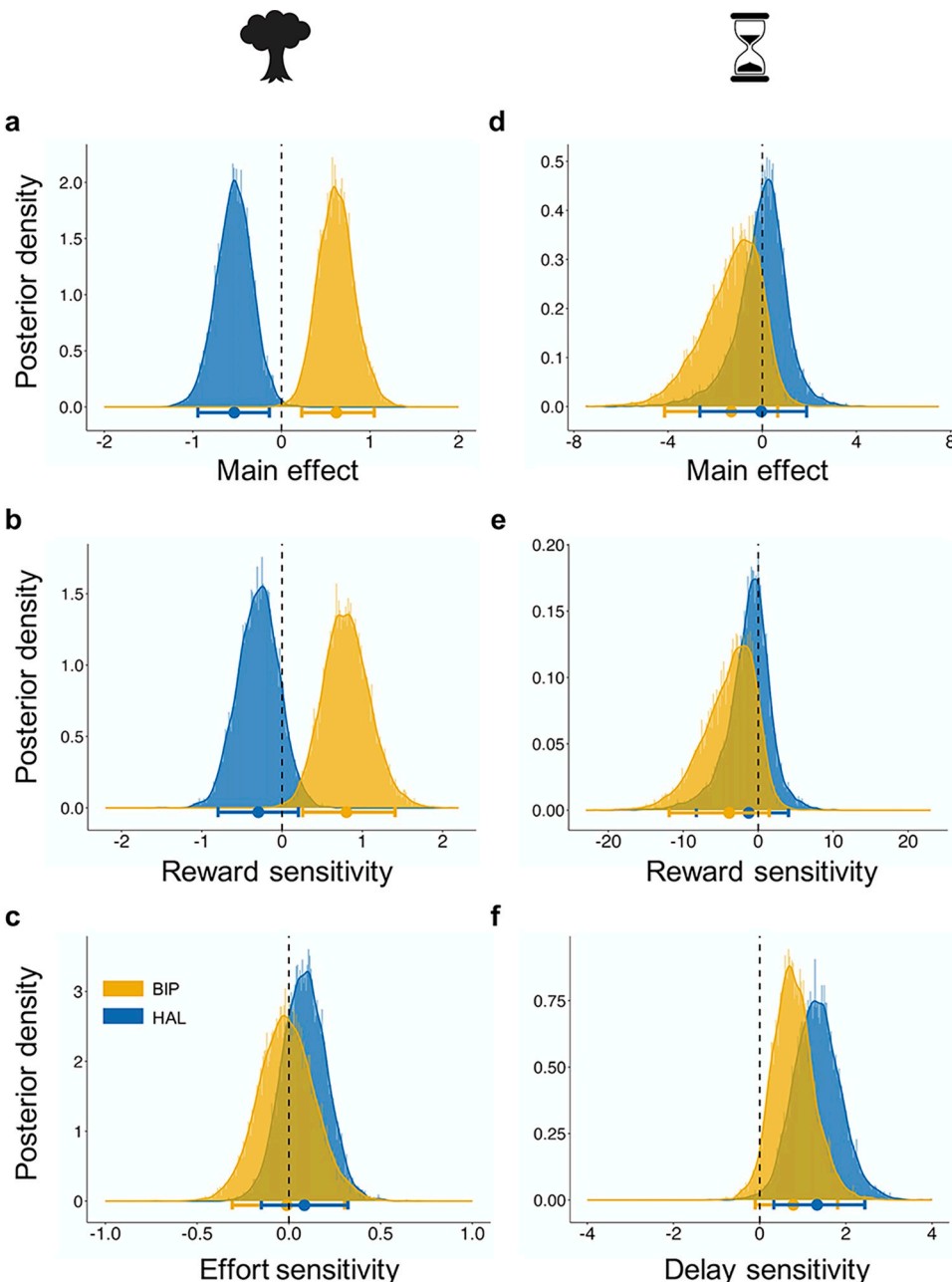

**Fig 3. Drug effects on choice behaviour.** Posterior distributions and 95% HDI of the logistic Bayesian GLMMs depict the estimate of each effect on choosing the high-cost option. (**a**) Biperiden credibly increased the overall willingness to invest physical effort for a corresponding reward, while haloperidol had the opposite effect. (**b**) Biperiden increased reward sensitivity in the effort discounting task, as indicated by a biperiden-by-reward interaction, without a credible effect of haloperidol. (**c**) In contrast, neither drug affected effort sensitivity. (**d**) In the delay discounting task, the willingness to tolerate delays for rewards was not affected by either drug. (**e**) Likewise, neither drug modulated reward sensitivity in the delay discounting task. (**f**) However, haloperidol decreased delay sensitivity, with no credible effect of biperiden. Here, a positive estimate of the interaction effect between haloperidol and delay indicates a reduction of the negative parameter estimate associated with delay, suggesting an attenuation of the impact of delay on choice behaviour. Bold dots represent the mean group-level estimate of the posterior distribution. The horizontal bars represent the group-level 95% HDI. GLMM, generalized linear mixed model; HDI, highest density interval.

Next, to investigate session and drug-order effects, we included session and the two-way interaction between session and drug as additional predictors. These analyses confirmed a main effect of session only for effort-based choices ($HDI_{Mean}$ = 0.49, $HDI_{95\%}$ = [0.13; 0.85]), suggesting increased preference for high-effort options in later sessions, likely reflecting task familiarity. Crucially, the interaction effects between either drug and session were not credible in both tasks (S14 and S15 Tables). This indicates that neither dopaminergic nor cholinergic manipulations modulated learning effects across sessions. Moreover, this finding underscores that the observed drug-induced effects cannot be explained by drug order effects.

To summarise, regression-based results show that haloperidol reduced the overall propensity to choose the high-cost options in the effort domain. While haloperidol had no effect on the average rate of selecting the high-cost option in delay-based choices, it attenuated participants' sensitivity to delay costs. In contrast, biperiden increased the overall propensity of selecting the high-cost option in the effort discounting task, opposite to the effect of haloperidol. Consistent with this, biperiden also increased sensitivity to reward magnitudes during effort-based choices.

## Drug effects on computational parameters

To obtain a more detailed understanding of the mechanism underlying the drug effects on behaviour described above, we used hierarchical Bayesian modelling. To begin with, we determined the best-fitting discounting models for both effort and delay, comparing 4 commonly used models (linear, parabolic, hyperbolic, and exponential). In line with previous findings [2,34–37], model comparisons revealed that effort discounting was best described by a parabolic model, suggesting a greater impact of changes in high rather than low-effort levels. In contrast, delay discounting was best described by a hyperbolic model, indicating a greater impact of changes in low rather than high-delay levels (S3 Table). For each model, we first calculated the SVs of all choice options, based on participant-specific weighing of reward magnitude and associated costs (i.e., effort and delay levels). By introducing condition-specific shift parameters, we captured potential drug effects on the effort and delay discounting parameter $\kappa$ (denoted as $s\,\kappa_{HAL}$ for haloperidol and $s\,\kappa_{BIP}$ for biperiden), with positive/negative shift parameter values indicating increased/decreased effort and delay discounting, respectively. We then used a softmax function to transform the option values into choice probabilities, with choice stochasticity being modelled by an inverse temperature parameter $\beta$. Again, condition-specific shift parameters for haloperidol and biperiden captured potential drug effects on choice stochasticity ($s\,\beta_{HAL}$ for haloperidol and $s\,\beta_{BIP}$ for biperiden), with positive/negative shift parameters indicating more deterministic/more stochastic decision-making (see Materials and methods, and Supporting Materials and Methods in S1 Text). Model validation and parameter recovery confirmed that both models accurately captured key features of the choice data (see Supporting Results in S1 Text).

The results from the computational model of the effort discounting task align with and extend the regression-based results presented above. Specifically, again biperiden and haloperidol exerted opposing effects on both the discounting and inverse temperature parameter. Haloperidol increased effort discounting, while biperiden diminished it. Similarly, haloperidol induced more stochastic choices, while biperiden led to more deterministic decisions (Table 1). For both drug-specific effects on the effort discounting parameter $\kappa$, we acknowledge that, strictly speaking, the 95% HDI of $s\,\kappa_{BIP}$ ($HDI_{Mean}$ = −0.012, $HDI_{95\%}$ = [−0.000; 0.027]) and $s\,\kappa_{HAL}$ ($HDI_{Mean}$ = 0.013, $HDI_{95\%}$ = [−0.025; 0.000]) did overlap with zero, albeit to a very small extent (Fig 4A). Notably, the density that did not overlap with zero still accounted for more than 94% of the posterior distribution (94.8% HDI > 0 for haloperidol

**Table 1. Summary of the group-level parameter estimates for the effort discounting task, including the mean, SD, and the lower and upper bounds of the 95% HDI interval.**

| Parameter | Mean | SD | 2.5% | 97.5% |
| --- | --- | --- | --- | --- |
| $\kappa$ | 0.102 | 0.008 | 0.086 | 0.118 |
| $\beta$ | 0.667 | 0.032 | 0.605 | 0.732 |
| $s\,\kappa_{HAL}$ | 0.013 | 0.007 | −0.000 | 0.027 |
| $s\,\kappa_{BIP}$ | −0.012 | 0.006 | −0.025 | 0.000 |
| $s\,\beta_{HAL}$ | −0.089 | 0.038 | −0.164 | −0.010 |
| $s\,\beta_{BIP}$ | 0.127 | 0.043 | 0.048 | 0.219 |

HDI, highest density interval; SD, standard deviation.

and 94.4% HDI > 0 for biperiden). This provides strong evidence for a credible modulation of effort discounting by cholinergic M1 and dopaminergic D2 receptor manipulation, despite the slight overlap.

A similar pattern emerged in the analysis of the inverse temperature parameter, further supporting the partly opposing effects of these neurotransmitters. The HDIs for $s\,\beta_{HAL}$ ($HDI_{Mean}$ = −0.089, $HDI_{95\%}$ = [−0.164; −0.010]) and $s\,\beta_{BIP}$ ($HDI_{Mean}$ = 0.127, $HDI_{95\%}$ = [0.048; 0.219]) did not overlap with zero, supporting the modulatory effects of these drugs on choice stochasticity. Specifically, haloperidol increased choice stochasticity, while biperiden exhibited the opposite effect (Fig 4B).

For the delay discounting task, computational modelling revealed credible evidence only for a dopaminergic, but not cholinergic modulation of delay discounting (Fig 4D and Table 2). In line with the diminished delay sensitivity reported above, haloperidol reduced delay discounting, making participants more willing to wait for greater financial rewards with increasing levels of delay ($s\,\kappa_{HAL}$: $HDI_{Mean}$ = −0.630, $HDI_{95\%}$ = [−1.083; −0.237]). For biperiden, we did not observe any shift of the discounting parameter ($s\,\kappa_{BIP}$: $HDI_{Mean}$ = −0.125, $HDI_{95\%}$ = [−0.410; 0.148]). Next, we examined the effects of dopaminergic and muscarinic antagonism on choice stochasticity (Fig 4E). Biperiden credibly increased the choice stochasticity ($s\,\beta_{BIP}$: $HDI_{Mean}$ = −0.043, $HDI_{95\%}$ = [−0.085; −0.000]), while we did not find credible evidence for a modulatory effect of haloperidol, as the 95% HDI overlapped with zero ($s\,\beta_{HAL}$: $HDI_{Mean}$ = −0.055, $HDI_{95\%}$ = [−0.114; 0.006]).

Lastly, to test whether putatively confounding side effects of the pharmacological treatment account for our main effects of interest, we tested medication effects on several control measures, including mood (alertness, calmness, and contentedness) and basic physiological parameters (heart rate, systolic, and diastolic blood pressure), using Bayesian linear mixed models. In short, biperiden administration induced reductions in systolic blood pressure and heart rate at $T_1$ and at $T_2$. Moreover, relative to placebo, both biperiden and haloperidol caused decreases in subjective alertness ratings at $T_2$ (all $HDI_{95\%}$ < 0). $T_1$ represents the measurement taken before participants began the effort discounting task, and $T_2$ represents the measurement taken after they completed the delay discounting task. Importantly, neither the changes in blood pressure or heart rate, nor reductions in alertness showed credible correlations with any shift parameter that was credibly modulated by haloperidol or biperiden in both tasks. This implies that the drug-induced changes in behaviour were not linked to drug effects on alertness, heart rate, or blood pressure. More detailed information can be found in the Supporting Results in S1 Text.

Taken together, the results derived from hierarchical Bayesian modelling are consistent with findings obtained from the GLMMs. Diminishing dopaminergic D2 receptor activity

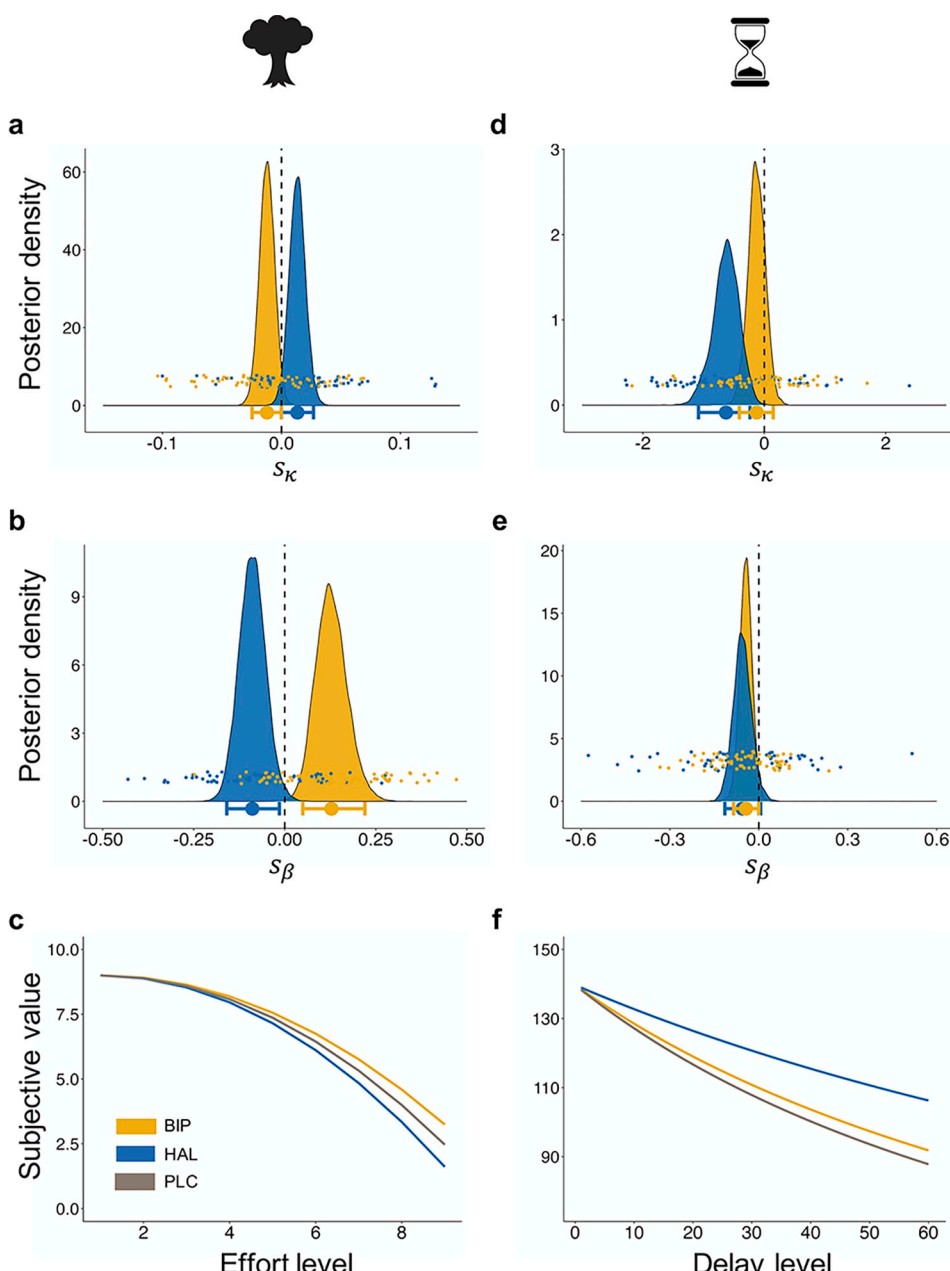

**Fig 4. Drug effect on computational parameters.** Posterior distributions and changes in the SV from the hierarchical Bayesian models. (**a**) In the effort discounting task, the discounting parameter κ is modulated in opposite directions by the drugs, with haloperidol increasing and biperiden decreasing effort discounting. (**b**) Similarly, these opposite effects are also present in the modulation of the softmax inverse temperature β, reflecting choice stochasticity. Biperiden administration led to more deterministic choices, while haloperidol induced more stochastic choices. (**c**) Modelled discounting functions show steeper discounting under haloperidol and flatter discounting under biperiden. (**d**) In the delay discounting task, the discounting parameter κ is reduced by haloperidol, with no credible modulatory effect of biperiden. (**e**) In contrast, the softmax inverse temperature β is reduced by biperiden, indicating more stochastic choices. (**f**) Overall, participants showed flatter discounting of future rewards under haloperidol compared to placebo. In (**a**), (**b**), (**d**), and (**e**), bold (light) dots represent the group-level (participant-level) mean estimate. The horizontal bars represent the group-level 95% HDI. In (**c**) and (**f**), the SVs are displayed as a discount function of effort and delay. Parabolic (effort discounting task) and hyperbolic (delay discounting task) functions are fitted on group-level mean estimates for each drug (see Materials and methods). HDI, highest density interval; SV, subjective value.

**Table 2. Summary of the group-level parameter estimates for the delay discounting task, including the mean, SD, and the lower and upper bounds of the 95% HDI interval.**

| Parameter | Mean | SD | 2.5% | 97.5% |
|---|---|---|---|---|
| $\kappa$ | −4.621 | 0.271 | −5.176 | −4.107 |
| $\beta$ | 0.356 | 0.045 | 0.272 | 0.449 |
| $s\,\kappa_{HAL}$ | −0.630 | 0.214 | −1.083 | −0.237 |
| $s\,\kappa_{BIP}$ | −0.125 | 0.142 | −0.410 | 0.148 |
| $s\,\beta_{HAL}$ | −0.055 | 0.030 | −0.114 | 0.006 |
| $s\,\beta_{BIP}$ | −0.043 | 0.021 | −0.085 | −0.000 |

HDI, highest density interval; SD, standard deviation.

decreases the willingness to invest effort for rewards, possibly by amplifying effort discounting. Conversely, the administration of a cholinergic M1 receptor antagonist produces opposite effects. However, this pattern of opponent dopaminergic and cholinergic effects on specific components of effort discounting was not present in the delay discounting task. Instead, the findings suggest a reduced impact of delay on decision-making under haloperidol, while biperiden affected only the choice stochasticity. This implies that contrasting effects of dopaminergic and cholinergic manipulations may reflect computationally specific, rather than universal effects.

## Across-task relationship of discounting behaviour

Lastly, as an exploratory analysis, we asked whether there was a relationship between discounting of delays versus efforts across individuals. In other words, we tested whether people showing steep discounting of physical efforts would also show steep delay-based discounting. In line with previous findings [36,38], we found no relationship between individuals' tendency to discount rewards based on efforts compared to delays (no credible correlation between the effort and delay discounting parameters $\kappa$ under placebo; $r = −0.17$, $HDI_{95\%} = [−0.06; 0.39]$).

## Discussion

In this study, we investigated the effects of pharmacologically manipulating dopaminergic and cholinergic neurotransmission on cost-benefit decision-making in healthy young adults. Specifically, we administered haloperidol or biperiden, 2 drugs that selectively block either dopamine D2-like or muscarinic M1 acetylcholine receptors, respectively, and tested the effects on discounting a monetary reward as a function of either effort or delay. In short, we found that reducing dopaminergic transmission at D2-like receptors decreased participants' willingness to invest physical effort for monetary rewards and attenuated the impact of delay on decision-making. In contrast, cholinergic M1 receptor manipulations promoted the preference for high-effort options, without affecting delay-based choices.

The goal of our study was 3-fold. First, we sought to conceptually replicate the boosting effects of dopamine on motivation. Second, we examined the contribution of acetylcholine, acting at M1 receptors, on cost-benefit decision-making, as the influence of muscarinic acetylcholine receptors on human (value-based) decision-making has rarely been reported. Third, we aimed to address the conflicting literature on dopamine's role in delay-based decision-making by providing data from a large sample using a within-subjects design.

Dopamine has been widely suggested to play a key role in various aspects of reward processing and cost-benefit decision-making [39–41]. It modulates choices based on expected rewards and costs associated with different options [42–44]. However, the direction of dopaminergic

manipulations on different aspects of costs remains ambiguous. Traditionally, dopamine has been implicated in promoting behaviour that maximises reward outcomes, suggesting that increased dopaminergic activity energises behaviour to approach high-reward options despite associated costs [7,45–47]. According to this view, increasing dopaminergic transmission should bias decision-making toward high-reward options, even when they involve higher costs such as effort, delay, and risk. Conversely, decreasing dopaminergic transmission should have the opposite effect.

In line with this view, in our study, the administration of a D2-like receptor antagonist did in fact reduce the willingness to invest physical effort for reward. However, contrary to the notion that dopamine supports reward-maximising behaviour, our findings in the delay discounting task did not confirm the expected pattern. We observed a decrease in delay discounting under haloperidol, rather than an increase. While there is indeed a substantial body of evidence suggesting enhancing effects of dopamine on motivation, as demonstrated by studies highlighting its involvement in promoting reward-seeking behaviour [8,9,48,49], the impact of pharmacological manipulations targeting other cost-related factors, such as delay [13,15,17,50] and risk [51–53], has produced rather inconsistent findings. These studies showed either contradicting results regarding dopaminergic manipulations or no effect at all. These inconsistencies challenge traditional views of dopamine and raise questions about the specific mechanisms through which dopamine modulates choices that incorporate time-dependent costs. A more recent theory proposes that dopamine, acting at D2-like receptors, biases action selection by increasing the preference for options with a proximity advantage over more distant alternatives [54,55]. This theory is based on studies in rodents, where firing rates of neurons in the nucleus accumbens are increased by spatially proximal rewards, promoting a decision bias towards nearby low-reward options [56]. This notion of proximity, originally referring to spatial proximity, has been extended to the context of psychological proximity [55]. According to this view, dopamine not only favours options that are physically closer in space, but also options that are psychologically closer, as in our case, rewards that are available sooner in time. Consequently, the administration of D2 receptor antagonists, which, according to this theory, are believed to reduce the proximity bias, have been shown in some studies to increase the preference for delayed and risky reward options, which lack the proximity advantage [19,20,51,57]. In other words, this theoretical framework proposes that dopamine may play a dual role in (1) promoting choices towards options with a psychological proximity advantage; and (2) weighing reward magnitudes against associated costs. Our findings of decreased delay discounting under haloperidol align with this theory, potentially suggesting a diminished preference for more proximal options when D2-receptor activity is reduced, resulting in increased endurance for higher (time) costs. On the other hand, the reduction of dopaminergic neurotransmission had the opposite effect on effort discounting, leading to a diminished willingness to bear higher (effort) costs. This apparently conflicting finding between effort and time costs may be explained by the fact that neither the low-effort nor the high-effort option had a pronounced psychological proximity advantage. Thus, in this specific context of cost-benefit decision-making, the proximity factor appears to be less relevant, and dopamine instead may promote behaviour aimed at maximising rewards simply by weighing rewards against associated costs. Importantly, our experimental paradigm did not directly manipulate psychological proximity, nor did it distinguish between proximity effects and the impact of delay. Therefore, specific effects of dopamine on psychological proximity in human delay discounting remain highly speculative and warrant further investigation with experimental paradigms specifically tailored to this question.

Low doses of dopamine receptor antagonists may facilitate dopaminergic transmission by primarily blocking presynaptic autoreceptors, which may increase rather than decrease

dopamine release [58,59]. We would however argue that such a presynaptic mechanism of action is unlikely to explain our findings. First, a positron emission tomography study found that the same dose of haloperidol as used here led to high levels of D2 receptor occupancy in the striatum [60]. Second, our pattern of results aligns with studies in rodents showing motivational deficits following ventral striatal dopamine depletion [61–63]. Third, high doses of haloperidol have been shown to reduce alertness [64], whereas drugs like methylphenidate that increase synaptic dopamine levels, enhance subjective ratings of alertness [65,66]. Consistent with this, we observed a reduction in alertness ratings following haloperidol administration. Together, this suggests that our results are best explained by a blockade of postsynaptic D2 receptors by haloperidol.

Notably, prior research has highlighted the contribution of other neurotransmitters such as serotonin [22,67], adenosine [68,69], and acetylcholine [23,29,70], in the weighing of costs and rewards. However, the exact role of these neurotransmitters in reward processing and cost-benefit decision-making in humans has been rarely investigated. Understanding the involvement of acetylcholine, due to its reciprocal activity with dopamine in the striatum, is of particular interest [24,26,27,71]. At the functional level, these mutual interactions are evident from the fact that muscarinic receptor antagonists diminish the extrapyramidal side effects of dopamine antagonists, and, vice versa, muscarinic agonists display antipsychotic properties, resembling the action of D2 antagonists [72,73]. Additionally, blocking M1 receptors has been shown to reverse motivational impairments induced by dopaminergic antagonism, further emphasising its potential role in modulating dopamine-related processes [29]. At the cellular level, it has been shown that M1 receptor activation inhibits D2-receptor–mediated effects in the striatum [74]. These observations highlight an interplay between acetylcholine and dopamine signalling. Therefore, in addition to testing dopaminergic D2-like receptor manipulations, we also investigated the role of cholinergic M1 receptor neurotransmission on decision-making.

Indeed, we observed opposing effects of dopaminergic and cholinergic manipulations within specific components of effort-, but not delay-based decision-making. Specifically, we found evidence for biperiden increasing the willingness to invest effort for rewards and, in line with this, decreasing effort discounting. In contrast, haloperidol reduced the general willingness to invest effort and increased effort discounting. However, we also observed drug effects that were not in opposite directions between biperiden and haloperidol. For example, biperiden increased reward sensitivity, while haloperidol had no credible effect on the impact of rewards on effort-based choices. Furthermore, haloperidol modulated decision times in both experimental tasks, whereas biperiden did not affect them. These findings suggest partially opposing effects between both neurotransmitters, mostly evident within specific components of effort-based choices.

This partially opposing mechanistic relationship between dopaminergic and cholinergic neurotransmission during the effort discounting task is further reflected in changes in choice stochasticity. Previous studies have linked D2 receptor antagonism with increased choice stochasticity [75,76]. Consistent with these findings, our results revealed that haloperidol administration increased stochasticity and thereby reduced value dependency on choices, while biperiden had the opposite effect. After lowering cholinergic M1 receptor activity, participants were more likely to choose the option with the highest expected value compared to placebo. However, it is important to note that biperiden had the opposite effect in the delay discounting task, increasing, rather than reducing choice stochasticity. This discrepancy indicates that the role of cholinergic neurotransmission in balancing deterministic versus stochastic behaviour is more complex and needs further investigation.

Some limitations should be noted. First, our focus was primarily on striatal D2 receptor activity, as haloperidol predominantly targets D2 receptors in the striatum [77]. However, it is

 

important to note that in the context of cost-benefit decision-making, dissociable roles of D1 versus D2 receptor activity have been reported [41,54,78], and thus the general role of dopamine beyond its selective activity on striatal D2 receptors remains unclear. Conversely, biperiden primarily targets M1 receptors in the cortex and striatum [79], making it challenging to determine the precise mechanisms underlying cortical and striatal cholinergic modulation and the reciprocal effects on dopaminergic activity. Second, previous research has suggested a U-shaped dose-response function for dopamine, indicating deleterious effects of both extremely high and extremely low levels of dopamine [80,81]. According to this idea, the same dopamine agent can produce opposing effects in different individuals. Therefore, it may be insightful to consider individual differences in baseline dopamine levels when studying the effects of dopaminergic manipulations. A recent study found evidence for the absence of a correlation between dopamine synthesis capacity and putative behavioural proxies of dopamine, such as working memory or trait impulsivity [82]. Consequently, investigating baseline dopamine levels require more costly and invasive techniques, such as positron emission tomography. Third, as participants were required to exert physical effort on each trial, fatigue effects could have developed in the effort discounting task. While our additional analysis confirmed a general fatigue effect and a dual role of haloperidol in both reducing the overall propensity to invest effort and in exacerbating the fatigue effect throughout the task, we acknowledge that recent studies revealed the existence of distinct states of fatigue [83,84]. Importantly, this was discovered by using task paradigms and computational models that were designed to distinguish between these different types of fatigue. Additionally, it is important to consider that potential motor effects, particularly in the context of dopaminergic manipulations, could also affect effort discounting behaviour. Future studies could extend our approach by incorporating measurements of force pulses to investigate potential motor-related effects and apply task designs and computational models specifically tailored to capture how drug manipulations might affect different forms of fatigue. Lastly, it is important to note that several factors preclude a direct quantitative comparison between the delay and effort discounting tasks. These include the distinct nature of rewards and costs (hypothetical versus real), varying reward magnitudes in both tasks (large versus small), and differences in task structure (fixed versus variable alternative option). Additionally, both tasks were consistently performed in a fixed order within and between participants (effort followed by delay). While this ensured consistent drug levels within each task, it is potentially introducing task order effects and possibly leading to differential drug concentrations between tasks. These methodological differences limit our ability to directly compare drug effects across the 2 cost domains. Future research would benefit from applying experimental paradigms that manipulate both delay and effort costs within the same task, allowing a more controlled and direct comparison of how pharmacological manipulations differentially influence sensitivity to these distinct cost types.

In conclusion, our findings support prior research indicating an invigorating effect of dopamine on motivation in an effort-based decision-making task. Moreover, our study contributed to understanding dopamine's involvement in temporal cost-benefit tradeoffs by revealing decreased delay discounting following dopaminergic D2-like receptor antagonism. Further, we demonstrate that the administration of biperiden, a muscarinic M1 receptor antagonist, had contrasting effects to those of dopaminergic D2 receptor antagonism in the general willingness to choose high-cost options and the effort discounting parameter. This suggests that in the context of human cost-benefit decision-making, the previously reported reciprocal relationship between both neurotransmitters may be limited only to specific components of behaviour. Our findings indicate that, while D2 receptor activity plays a role in integrating both delay and effort costs, acetylcholine, acting at M1 receptors, may have a more specific role in effort processing.

## Materials and methods

### Ethics

Ethical approval for the study (2021–1549) was obtained from the Ethics Committee of the Medical Faculty of the University of Düsseldorf, Germany. The study was conducted in accordance with the principles expressed in the Declaration of Helsinki. Prior to participating, all volunteers provided informed written consent.

### Participants

Participants were recruited for the study via online advertisements and university postings. They were initially screened via email interviews to verify compliance with the inclusion and exclusion criteria. Inclusion criteria included an age range of 18 to 35 years, a body weight ranging from 60 to 90 kg, and a body mass index equal to or greater than 18 and less than 28. Exclusion criteria included a history of psychiatric or neurological illnesses, current intake of prescription medication (excluding oral contraceptives), current pregnancy or breastfeeding, and the presence of any medical conditions contraindicated for the drugs used in the study. Additionally, participants with a history of drug use were excluded, with exceptions for alcohol, nicotine, and cannabis, which were limited to consumption of less than 14 units of alcohol per week, less than 5 cigarettes per day, and no cannabis consumption within the past month.

From an initial group of 96 individuals attending the medical screening on site, 33 candidates decided not to proceed, or they were excluded due to not meeting the participation criteria (despite otherwise indicated in the prior interview). Eventually, a total of 63 volunteers were enrolled in the study. One participant had to be excluded due to experiencing side effects following the biperiden testing session, resulting in a final sample size of 62 participants (32 female; mean age (standard deviation, SD): 22.79 (3.20); age range: 18 to 35). The sample did not include any nonbinary participants. All participants received a fixed reimbursement of 240 € for their participation in all testing sessions and were provided with a flexible payment based on their performance in the tasks.

### Procedure

A double-blind, randomised, within-subject design was employed. Participation involved 3 testing sessions, each separated by a minimum of 1 week to ensure complete drug washout. Prior to participation, on a separate day, participants underwent a medical screening session to determine their eligibility. This screening session consisted of a clinical interview, medical assessment, practice trials of both tasks, and completion of several questionnaires assessing personality traits, including Apathy Evaluation Scale (AES) [85,86], Beck's Depression Inventory (BDI) [87,88], and Barratt Impulsiveness Scale (BIS-15) [89,90]. The final decision regarding participant suitability for the study was made by a physician at the end of the screening session.

Each testing day followed the same procedure, with the only difference being the administration of a placebo, biperiden (4 mg), or haloperidol (2 mg) in separate sessions. Testing sessions were scheduled between 9 AM and 12 PM, with efforts made to assign participants to the same time slot consistently. Participants were instructed to fast overnight and consume only water before the testing session. Additionally, female participants completed a pregnancy test before the start of each testing day. Upon arrival, participants underwent a screening conducted by a physician, who restated the potential effects of the drug and provided a recap of the testing day's procedures. Subsequently, participants received the assigned drug treatment along with a standardised breakfast. Orally administered biperiden reaches its peak

concentration between 1 and 1.5 h after administration [72,91], while haloperidol reaches its peak concentration only between 2 and 6 h after administration [92]. In order to follow a double-blind procedure while also accounting for variations in the drugs' peak times, a dummy drug application was included. Thus, on each testing day, a second capsule was administered 120 min after the first drug application, resulting in the planned delay between drug and effort discounting task of approximately 180 min for haloperidol (179.57 ± 2.61) and, on a separate day, approximately 60 min (59.65 ± 1.30) for biperiden. The second task was performed 205.59 min (± 3.12) after haloperidol and 86.24 min (± 6.09) after biperiden administration. Shortly before starting the experimental task, participants further completed the trail-making test A [93]. Blood pressure and heart rate were measured 3 times throughout the testing day: at $T_0$ (before drug administration), $T_1$ (before starting the effort discounting task), and $T_2$ (after finishing the delay discounting task). These measurements coincided with mood assessments using Visual Analogue Scales [94]. Please refer to Fig 1A for a detailed explanation of the procedure. The tasks were presented in a fixed order during each session, starting with the effort discounting task (Fig 1B) and concluding with the delay discounting task (Fig 1C). These discounting paradigms were implemented using Visual Basic software and displayed on a 15.6" laptop screen (Dell Latitude E5550). Participants used an external keypad to indicate their choices, arranging the laptop and keypad in a manner that was most comfortable for them.

## Experimental tasks

**Effort discounting task.** Participants engaged in a modified version of the Apple Tree Task, which has been previously utilised with both healthy volunteers and patients with neurological diseases [5,95,96]. Unlike the original task where participants accepted or rejected offers, in this modified task, they were presented with 2 alternative options on each trial. Prior to each experimental session, the participants' maximum voluntary contraction (MVC) was assessed by having them grip a handheld dynamometer (Vernier, Orlando, United States of America) with their dominant hand as forcefully as possible. The MVC was determined immediately before starting the task by measuring the highest force exerted over 3 contractions.

During each trial, they had to choose between a high-reward/high-effort (high-cost) and a low-reward/low-effort (low-cost) offer. Participants had unlimited time for their decision. Notably, both the reward levels and effort levels of each option were varied among 5 possible levels. The reward levels ranged from 2 to 16 (2, 4, 8, 12, and 16), and the effort levels ranged from no effort to 80% (0%, 20%, 40%, 60%, and 80%) of the individually determined MVC. After making a choice, participants were given a 5-s window to squeeze the handheld dynamometer and reach the required effort level. They had to maintain the required force for at least 1 s. Throughout this effort production period, a bar visually represented the force exerted, providing real-time feedback. Following successful trials, participants received feedback on the reward earned during the trial. If participants failed to reach the designated effort level, no apples were gathered. In cases where participants chose an offer that required no effort, they had to wait for the same duration without engaging in squeezing. At the end of the experiment, we asked participants to rate the perceived level of demand for each effort level using a Likert scale ranging from 0 ("not demanding at all") to 20 ("extremely demanding"). This subjective rating allowed us to examine whether the administration of the drugs had any effects on participants' perception of the demand associated with each effort level.

Each participant completed a total of 125 trials, divided into 5 blocks. The trial structure was full randomised for each participant and experimental session. To prevent strategic behaviour and mitigate delay discounting effects, all blocks and trials were the same length, regardless of previous choices made. Importantly, patients were required to squeeze the handheld

dynamometer after every trial (if effort production was chosen), meaning that no hypothetical choices were made. Based on their task performance, participants received a flexible payment consisting of 1 cent for each apple they collected.

**Delay discounting task.**   After completing the effort discounting task, participants proceeded to the delay-based decision-making task. In this task, they were presented with a similar choice paradigm, but this time between a high-reward/high-delay (high-cost) and a low-reward/low-delay (low-cost) option on each trial, again without any time limit. The tasks shared a common structure, with participants consistently facing decisions involving a tradeoff between a more favourable outcome with higher associated costs and a less favourable outcome with fewer costs. In contrast to the previous task, the low-cost option was fixed at 20 € and available immediately across all trials. The high-cost option varied between 20.20 € and 260 € (20.20, 20.40, 21, 22, 24, 30, 36, 40, 50, 60, 80, 100, 140, 200, and 260 €), with associated delays ranging from 1 day to 60 days (1, 2, 3, 5, 8, 30, and 60). This resulted in a total of 105 unique high-cost combinations.

Participants completed 2 blocks, with each block containing all unique combinations, resulting in a total of 210 trials. We used a pseudorandomised trial order, counterbalanced across experimental sessions but not across participants. Consistent with previous studies, all choices in this task were hypothetical [97,98]. However, participants were instructed to imagine that one of their choices would be randomly selected and paid out.

## Statistical analysis

**Regression-based analysis.**   To investigate the influence of drug administration (compared to placebo) and experimental manipulations (i.e., reward magnitude, delay, and/or effort) on choice behaviour, we employed logistic Bayesian GLMMs, using the brms package [99] in R (Version 4.1.3). In these models, we regressed choices (choosing the high-cost option versus choosing the low-cost option) on fixed-effect predictors including drug, reward, cost (i.e., delay or effort), and all possible interaction terms. Importantly, to test drug-specific effects on behaviour, we included the drug condition (placebo, haloperidol, biperiden) as a 3-level categorical predictor with placebo set as the reference. Thus, all drug-related main and interaction effects are estimated in reference to the placebo. For the effort discounting task, as predictors we used the difference term between both the reward magnitude and effort levels of the 2 presented options (high-cost option–low-cost option). In the delay discounting task, the reward and delay levels of the varying high-cost option were included as regressors. To account for individual differences, all fixed-effect predictors were also modelled as random slopes in addition to subject-specific random intercepts. To ensure robust and informative analyses, we followed the approach suggested by Gelman and colleagues [100]. Weakly informative priors were employed, with nonbinary variables scaled to have a mean of 0 and a standard deviation of 0.5. Posterior distributions of the parameter estimates were obtained by running 4 chains with 3,000 samples (1,000 samples for warmup). We present the 95% highest density intervals (HDIs) of the estimates to capture the uncertainty in the parameter estimates. The 95% HDI indicates that there is a 95% probability that the true parameter value falls within this interval. If the 95% HDI does not overlap with zero, it provides credible evidence that the respective model parameter is meaningful [101,102]. For detailed information on the models employed in this analysis, please refer to the Supporting Materials and Methods in S1 Text.

**Order and fatigue control analysis.**   To further test for potential confounding effects on choice behaviour, we ran separate control regression analyses. For these models, we used the structure of the original GLMMs but included additional fixed-effect predictors to control for session and fatigue effects. Specifically, we included trial number as a predictor to account for

 

potential systematic changes in choice behaviour throughout the task progress. In separate models, session was included as a fixed effect to control for potential learning or drug order effects across different experimental sessions. Importantly, to capture potential interactions between these control variables and the drug manipulations, we included the two-way interaction terms in the fixed-effects structure. This allows to investigate whether the effects of the drug manipulation on choice behaviour differed as a function of the task progress or experimental session. All other aspects of the model specification, including random-effects structure, priors, and estimation procedures, remained identical to the original regression models described above.

**Hierarchical Bayesian modelling.** Next, to investigate the impact of dopaminergic and cholinergic manipulations on underlying cognitive processes, specifically how individuals integrate rewards and costs to discount subjective reward values, we employed hierarchical Bayesian modelling. This approach allowed us to estimate both group-level hyperparameters and individual subject-level estimates, leveraging the hierarchical structure of the experimental design. By incorporating information from each individual's estimates into the group estimates and vice versa, we obtained more robust and reliable parameter estimates compared to conventional methods like maximum likelihood estimation [101].

To capture how changes in the SVs are influenced by reward, effort, and delay, we employed a single-parameter discounting model to determine which discounting function best describes the observed behaviour. Initially, participants' responses from the placebo condition were fitted to 4 commonly used models for discounting: linear, parabolic, hyperbolic, and exponential [2,36,103,104]. Model fits were compared using the trial-based leave-one-out information criterion (LOOIC) from the loo package [105].

Consistent with previous findings, delay discounting was best described by a hyperbolic model [34,35], whereas effort discounting showed the best fit with a parabolic model [2,36,37]. See Supporting Materials and Methods in S1 Text for more details. The discounting parameter in the delay discounting model was modelled in log space due to the skewed distribution of the values towards zero.

$$SV\left(HC_{(t)}\right) = \frac{R_{(t)}}{1 + \exp(\kappa) * D_{(t)}} \tag{1}$$

$$SV(HC_{(t)}) = R_{HC_{(t)}} - \kappa * E_{HC_{(t)}}^{2} \tag{2}$$

$$SV(LC_{(t)}) = R_{LC_{(t)}} - \kappa * E_{LC_{(t)}}^{2} \tag{3}$$

In the delay discounting model (Eq 1), $R_t$ represents the reward magnitude and $D_t$ represents the delay in days of the high-cost option (HC) on trial t, while the SV of the low-cost option (LC) was fixed at 20 for each trial. In contrast, the effort discounting models (Eqs 2 and 3) involve varying levels of both options. In this model, the SV of the HC option and the SV of the LC option are calculated separately. Once again, $R_t$ denotes the corresponding reward magnitude, and $E_t$ represents the associated effort in percentage on trial t.

$$\kappa = \kappa_{PLC} + I_{HAL_{(t)}} * s_{\kappa_{HAL}} + I_{BIP_{(t)}} * s_{\kappa_{BIP}} \tag{4}$$

Consistent with previous studies that examined drug effects on changes in discounting [20,106,107], we extended the original single-parameter model by incorporating 2 additional free parameters (Eq 4). In both cases, κ is the discounting parameter, either reflecting delay discounting or effort discounting. A higher κ value indicates a greater degree of discounting,

 

whereas a lower κ value suggests less discounting. To capture potential drug effects (compared to placebo), 2 separate shift parameters ($s\,\kappa_{HAL}$ for haloperidol and $s\,\kappa_{BIP}$ for biperiden) were included to model changes in the discounting rate. A positive shift parameter indicates that the corresponding drug increases discounting, while a negative shift parameter suggests that the drug decreases discounting. The condition for each trial is indicated by the dummy-coded variable $I$, which indicates the drug condition of the current trial.

$$P\left(HC_{(t)}\right) = \frac{\exp(SV(HC_{(t)})*\beta)}{\exp(SV(HC_{(t)})*\beta) + \exp(SV(LC_{(t)})*\beta)} \tag{5}$$

$$\beta = \beta_{PLC} + I_{HAL_{(t)}}*s_{\beta_{HAL}} + I_{BIP_{(t)}}*s_{\beta_{BIP}} \tag{6}$$

We then used a softmax function to transform the option values to choice probabilities (Eq 5). The choice stochasticity was modelled using the inverse temperature parameter β. A lower β value indicates more stochastic behaviour. Conversely, a higher β value suggests more deterministic choices.

Similar to the previous analysis, 2 additional shift parameters ($s\,\beta_{HAL}$ for haloperidol and $s\,\beta_{BIP}$ for biperiden) were introduced to capture potential drug effects on choice stochasticity (Eq 6). A positive $s\,\beta$ parameter indicates that the corresponding drug decreases the level of stochasticity in choices, while a negative $s\,\beta$ parameter suggests that the drug increases stochasticity.

Model estimation was performed using MCMC sampling as implemented in STAN [108] via R and the rSTAN package (Version 2.21.0). We utilised separate group-level distributions for all parameters in the placebo condition (i.e., κ and β), as well as for the shift parameters (i.e., $s\,\kappa_{HAL}$, $s\,\kappa_{BIP}$, $s\,\beta_{HAL}$, and $s\,\beta_{BIP}$), which capture potential modulatory effects of the drugs. Prior distributions for the parameter means and standard deviations were chosen within plausible ranges based on previous findings [20,106,109–111]. The sampling process involved running 4 chains with 4,000 iterations after a warmup period of 3,000 iterations. Chain convergence was assessed using the Gelman-Rubinstein convergence diagnostic r-hat, with values less than 1.01 considered acceptable [112]. We report the mean of the posterior group distribution for all parameters, along with the associated 95% HDI.

Please refer to the Supporting Materials and Methods in S1 Text for details of the prior distributions, model comparison, and model estimation procedure.

**Model validation and parameter recovery.** To assess the model's ability to capture and recover important characteristics of the data, we conducted a model validation and parameter recovery analysis. This involved generating 500 synthetic datasets per participant using the posterior distributions of subject-level parameters obtained from the winning models. From these 500 datasets, we randomly selected 10 datasets for each participant and conducted the same analysis as described above, using the synthetic datasets instead of the actual data.

First, to validate that the simulated data accurately captures the key features of the participants' behaviour, we simulated the relative choice rates and visualised how behavioural patterns changed as a function of varying levels of costs and rewards. Next, to evaluate the recovery of group-level parameters, we examined whether the simulated group-level parameters fell within the 95% HDI of the actual group-level parameter distribution. For the subject-level parameter estimates, we calculated the correlation between the simulated (averaged across all 10 simulated datasets) and true estimated subject-level parameters. The results of the model validation and parameter recovery are presented in the Supporting Results in S1 Text.

**Drug effects on vital signs, mood, trail-making performance, MVC, and effort rating.** The effects of the drugs on mood (alertness, calmness, and contentedness) and physiological

measures (heart rate, systolic, and diastolic blood pressure) were analysed using Bayesian linear mixed models. If credible drug interactions were found, we further explored the relationship between drug-induced alterations in behaviour and potential explanatory factors, including changes in subjective mood ratings and/or physiological responses. Specifically, we calculated the absolute differences in mood ratings and physiological parameters between $T_0$ (before drug administration) and the time point where a credible drug effect was observed (either $T_1$ or $T_2$). These difference values were then correlated with the shift parameters that capture drug-induced changes in performance. The purpose of this Bayesian correlation analysis was to exclude that drug-induced changes in mood ratings or physiological measures could account for the observed drug-induced changes in behaviour. We further investigated potential confounding drug effects on trail-making response times, MVC, and subjective effort perception. These analyses did not reveal significant effects of drug manipulation on any of these measures (see Supporting Results in S1 Text). More detailed descriptions of these analysis are presented in the Supporting Materials and Methods in S1 Text.

**Association between task- and drug-specific computational parameter estimates.** We investigated the association between discounting tendencies across different cost domains, specifically exploring whether participants who exhibit stronger effort discounting also displayed stronger delay discounting. To examine this, we performed a Bayesian correlation analysis using the mean estimates of the κ parameters from the placebo conditions in both the effort and delay discounting task.

**Computational parameter estimates, self-ratings scores, and demographics.** In a supplementary analysis, we investigated potential associations between self-reported questionnaire scores, demographic data (including age and sex of the participants), and the estimated discounting parameters κ from the baseline (placebo) condition of both tasks. To this end, we performed robust linear regressions with the respective discounting parameter as the outcome variable (see Supporting Materials and Methods and Supporting Results in S1 Text).

## Supporting information

**S1 Text. Supporting Results; Supporting Materials and Methods.**
(DOCX)

**S1 Data. Excel file containing the underlying numerical data for the effort discounting task in Figs 2 and S8.**
(XLSX)

**S2 Data. Excel file containing the underlying numerical data for the delay discounting task in Figs 2 and S8.**
(XLSX)

**S1 Fig. Choice behaviour in the placebo (baseline) condition.** Posterior distributions and 95% HDI of the logistic Bayesian generalized linear mixed models depict the estimate of each task parameter on choosing the high-cost option. (**a**) Higher reward magnitudes increased the overall willingness to invest physical effort for a corresponding reward in the effort discounting task. (**b**) Higher levels of effort had the opposite effect. (**c**) Similarly, higher reward magnitudes increased the likelihood to choose the high-cost option in the delay discounting task. (**d**) In contrast, higher levels of delay decreased the willingness to choose the high-reward/high-delay option. Bold dots represent the mean group-level estimate of the posterior distribution. The horizontal bars represent the group-level 95% highest density interval.
(TIF)

**S2 Fig. Model validation for the effort (a–c) and delay discounting task (d–f).** Plots depict the averaged overall proportion of choosing the high-cost option as a function of reward and cost for both the effort (**a–c**) and the delay (**d–f**) discounting task. The upper panels display the actual data and the lower panels present the simulated data for comparison. Group-level means are indicated by dots, with error bars representing the standard error of the mean. In the effort discounting task, reward levels are presented as the difference in magnitude between the high- and low-cost option, and in the delay discounting task, reward levels are shown as the reward value of the high-cost option. Likewise, the effort level corresponds to the difference between the proportions of the individually calibrated MVC of the high- and low-cost option, while the delay levels indicate the delay of the high-cost option.
(TIF)

**S3 Fig. Parameter recovery of the group-level posterior distributions for the effort (a–f) and delay discounting task (g–l).** We tested the ability of our models to recover parameters using simulated datasets. Each panel displays the actual distribution (red) of each parameter of interest alongside 10 corresponding simulated datasets (blue). Horizontal bars, depicted in red, represent the group-level 95% HDI of the actual parameter estimate, with dots indicating the mean of the distribution. The shaded area in blue depicts the 95% HDI of the simulated parameter estimates. We evaluated whether the mean estimates of the simulated group-level parameter values fell within the 95% HDI of the true parameter distribution. The parameter recovery analysis of the group-level distribution demonstrated positive results, as all mean parameter estimates of the simulated data are located within the 95% HDI of the actual dataset.
(TIF)

**S4 Fig. Parameter recovery of the subject-level parameters for the effort (a–f) and delay discounting task (g–l).** We calculated the Pearson correlation coefficients between the mean subject-level posterior distribution of the simulated and actual data. For the simulated data, subject-level means were averaged across each dataset. The correlation coefficients demonstrate strong to excellent correlations (all $r > 0.8$), further confirming that the models are able to accurately to recover the actual task parameter values.
(TIF)

**S5 Fig. Physiological effects of drug administration.** Plots depict the change in each physiological parameter following drug administration. (**a**) Diastolic BP decreased at T1 ($HDI_{Mean} = -3.75$, $HDI_{95\%} = [-5.71; -1.85]$) and T2 ($HDI_{Mean} = -2.99$, $HDI_{95\%} = [-4.96; -1.07]$) compared to T0. However, we did not find a credible main or interaction effect of drug, implying that diastolic BP decreases as the task progresses irrespective of drug administration. (**b**) Systolic BP reflected notable two-way biperiden interactions at T1 and T2 (Biperiden x T1: $HDI_{Mean} = -3.91$, $HDI_{95\%} = [-7.74; -1.07]$; Biperiden x T2: $HDI_{Mean} = -4.80$, $HDI_{95\%} = [-8.74; -0.95]$), suggesting a more pronounced decrease in systolic BP following Biperiden application. (**c**) Heart rate was reduced at T1 ($HDI_{Mean} = -6.82$, $HDI_{95\%} = [-10.74; -2.97]$) and T2 ($HDI_{Mean} = -7.88$, $HDI_{95\%} = [-11.83; -3.89]$) relative to T0. A credible two-way interaction was found between Biperiden at T1 and T2 (Biperiden x T1: $HDI_{Mean} = -6.82$, $HDI_{95\%} = [-10.74; -2.97]$; Biperiden x T2: $HDI_{Mean} = -7.88$, $HDI_{95\%} = [-11.83; -3.89]$), indicating that, analogously to the drop in systolic BP, the drop in heart rate at T1 and T2 is more pronounced following biperiden administration compared to placebo. Dots represent the group-level mean, error bars depict the standard error of the mean.
(TIF)

**S6 Fig. Subjective mood rating effects of drug administration.** Plots show alterations in subjective mood ratings following drug administration. No credible effects of time or drug were

found for (**a**) contentedness and (**b**) calmness ratings. However, (**c**) alertness ratings notably decreased over time, exhibiting a credible drop at both T1 ($HDI_{Mean} = -0.08$, $HDI_{95\%} = [-0.16; -0.01]$) and T2 ($HDI_{Mean} = -0.23$, $HDI_{95\%} = [-0.30; -0.15]$) compared to T0. Importantly, at T2, credible two-way interaction effects were found for biperiden and haloperidol (Biperiden x T2: $HDI_{Mean} = -0.15$, $HDI_{95\%} = [-0.26; -0.05]$; Haloperidol x T2: $HDI_{Mean} = -0.15$, $HDI_{95\%} = [-0.25; -0.04]$), suggesting that both drugs led to more pronounced reductions in alertness ratings towards the end of the experiment, compared to placebo. Dots represent the group-level mean, error bars depict the standard error of the mean.
(TIF)

**S7 Fig. Drug effects on trail-making test performance, subjective effort ratings, and MVC.** (**a**) No credible drug effects were observed for response times during trail-making test A. However, credible main effects of session were found (Session 2: $HDI_{Mean} = -4.77$, $HDI_{95\%} = [-8.41; -1.17]$; Session 3: $HDI_{Mean} = -6.40$, $HDI_{95\%} = [-10.14; -2.66]$), suggesting that participants became faster after the first session, possibly due to familiarity with the task. (**b**) Effort ratings were modulated only by increasing effort levels ($HDI_{Mean} = 5.15$, $HDI_{95\%} = [4.81; 5.49]$), suggesting no drug effects on the subjective experience of effort demand. (**c**) MVC was not credibly modulated by drug administration, indicating that participant's ability to exert effort was not modulated by drug either. Dots represent the group-level mean and error bars represent standard error of the mean.
(TIF)

**S8 Fig. Decision times in milliseconds (ms) in the effort (left panel) and delay (right panel) discounting tasks**. (**a**) Overall decision times in the effort discounting task (left) were not affected by haloperidol or biperiden. However, in the delay discounting task (right), haloperidol reduced the decision times. (**b**) Decision times decreased with larger rewards. Haloperidol reduced this speed-up effect in both the effort (left) and delay discounting task (right panel). (**c**) Conversely, decision times increased with higher cost levels. This effect was not modulated by any drug in the effort discounting task (left). However, in the delay discounting task (right), haloperidol diminished the decelerating effect of increasing delay levels. (**a**) Shows group-level (single-subject) means represented by bold (light) dots. (**b** and **c**) Display averaged group-level means per reward and cost level, with error bars representing the standard error of the mean. Reward levels are presented as the difference in magnitude between the high- and low-cost option in the effort discontinuing task and as the absolute reward value of the high-cost option in the delay discounting task. Likewise, the effort level represents the difference between the proportions of the individually calibrated MVC of the high- and low-cost option, while the delay level indicates the delay of the high-cost option. The data underlying the effort discounting task (left panel) can be found in S1 Data, and the data underlying the delay discounting task (right panel) can be found in S2 Data.
(TIF)

**S9 Fig. Drug effects on decision times.** Posterior distributions and 95% HDI of the Bayesian linear mixed models depict the estimate of each effect on decision times. (**a**) In the effort discounting task, overall decision times were not credibly influenced by either haloperidol or biperiden. (**b**) Notably, a haloperidol-by-reward interaction revealed a reduced reward sensitivity after haloperidol administration. (**c**) On the other hand, sensitivity towards increasing levels of effort is not affected by either drug. (**d**) In the delay discounting task, haloperidol credibly reduced overall decision times, while biperiden did not show any credible effect. (**e**) Analogous to the effort discounting task, a credible haloperidol-by-reward interaction demonstrate reduced reward sensitivity following haloperidol administration. (**f**) Furthermore,

haloperidol administration credibly reduced delay sensitivity, while biperiden had no effect on the impact of delay on decision times. Bold dots represent the mean group-level estimate of the posterior distribution. The horizontal bars represent the group-level 95% HDI.
(TIF)

**S1 Table. Bayesian generalized linear mixed models of the effort discounting task, regressing choices (high-cost vs. low-cost option) on predictors for drug, reward (difference between high-cost vs. low-cost reward level), effort (difference between high-cost vs. low-cost effort level), and their interaction terms.**
(DOCX)

**S2 Table. Bayesian generalized linear mixed models of the delay discounting task, regressing choices (high-cost vs. low-cost option) on predictors for drug, reward (high-cost option reward), delay (high-cost option delay), and their interaction terms.**
(DOCX)

**S3 Table. Model comparison for the effort and delay discounting task. To compare the validity of each model, we used the leave-one-out cross-validation information criterion (LOOIC) procedure. A lower LOOIC score indicates a better-fitting model.**
(DOCX)

**S4 Table. Bayesian generalized linear mixed models of the effort discounting task–baseline session; regressing choices (high-cost vs. low-cost option) on predictors for reward (difference between high-cost vs. low-cost reward level), effort (difference between high-cost vs. low-cost effort level), and their interaction terms.**
(DOCX)

**S5 Table. Bayesian generalized linear mixed models of the delay discounting task–baseline session; regressing choices (high-cost vs. low-cost option) on predictors for reward (high-cost option reward), delay (high-cost option delay), and their interaction terms.**
(DOCX)

**S6 Table. Bayesian linear mixed models of the effort discounting task, regressing decision times on predictors for drug, reward (difference between high-cost vs. low-cost reward level), effort (difference between high-cost vs. low-cost effort level), and their interaction terms.**
(DOCX)

**S7 Table. Bayesian linear mixed models of the delay discounting task, regressing decision times on predictors for drug, reward (high-cost option reward), delay (high-cost option delay), and their interaction terms.**
(DOCX)

**S8 Table. Fixed effects from robust linear regression model with κ as dependent variable and questionnaire total scores, sex, and age as independent variable for the effort discounting task.**
(DOCX)

**S9 Table. Fixed effects from robust linear regression model with κ as dependent variable and questionnaire total scores, sex, and age as independent variable for the delay discounting task.**
(DOCX)

**S10 Table. Fixed effects from robust linear regression model with κ as dependent variable and questionnaire subscales as independent variable for the effort discounting task.** (DOCX)

**S11 Table. Fixed effects from robust linear regression model with κ as dependent variable and questionnaire subscales as independent variable for the delay discounting task.** (DOCX)

**S12 Table. Bayesian generalized linear mixed models of the effort discounting task–fatigue effects; regressing choices (high-cost vs. low-cost option) on predictors for drug, reward (difference between high-cost vs. low-cost reward level), effort (difference between high-cost vs. low-cost effort level), and their interaction terms, as well as trial number and two-way trial number × drug interactions.** (DOCX)

**S13 Table. Bayesian generalized linear mixed models of the delay discounting task–fatigue effects; regressing choices (high-cost vs. low-cost option) on predictors for drug, reward (high-cost option reward), delay (high-cost option delay), and their interaction terms, as well as trial number and two-way trial number × drug interactions.** (DOCX)

**S14 Table. Bayesian generalized linear mixed models of the effort discounting task–session effects; regressing choices (high-cost vs. low-cost option) on predictors for drug, reward (difference between high-cost vs. low-cost reward level), effort (difference between high-cost vs. low-cost effort level), and their interaction terms, as well as session and two-way session × drug interactions.** (DOCX)

**S15 Table. Bayesian generalized linear mixed models of the delay discounting task–session effects; regressing choices (high-cost vs. low-cost option) on predictors for drug, reward (high-cost option reward), delay (high-cost option delay), and their interaction terms, as well as session and two-way session × drug interactions.** (DOCX)

## Acknowledgments

We express our gratitude to the students who conducted and supported the data collection: Bibiana Ackers, Evelina Andronova, Alina Hansen, and Maria Angelina Morales Diaz. Computational infrastructure and support were provided by the Centre for Information and Media Technology at Heinrich Heine University Düsseldorf.

## Author Contributions

**Conceptualization:** Mani Erfanian Abdoust, Gerhard Jocham.

**Data curation:** Mani Erfanian Abdoust.

**Formal analysis:** Mani Erfanian Abdoust, Monja Isabel Froböse, Gerhard Jocham.

**Funding acquisition:** Gerhard Jocham.

**Investigation:** Mani Erfanian Abdoust, Elisabeth Schreivogel.

**Methodology:** Mani Erfanian Abdoust, Monja Isabel Froböse, Gerhard Jocham.

**Project administration:** Mani Erfanian Abdoust, Monja Isabel Froböse, Alfons Schnitzler, Gerhard Jocham.

**Resources:** Alfons Schnitzler, Elisabeth Schreivogel, Gerhard Jocham.

**Software:** Mani Erfanian Abdoust.

**Supervision:** Monja Isabel Froböse, Alfons Schnitzler, Gerhard Jocham.

**Validation:** Mani Erfanian Abdoust, Monja Isabel Froböse, Gerhard Jocham.

**Visualization:** Mani Erfanian Abdoust.

**Writing – original draft:** Mani Erfanian Abdoust.

**Writing – review & editing:** Monja Isabel Froböse, Alfons Schnitzler, Elisabeth Schreivogel, Gerhard Jocham.

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
