## [Editor Report · Decision Letter 0]

19 Dec 2023

Dear Dr Erfanian Abdoust, 

Thank you for submitting your manuscript entitled "Distinct roles of dopamine and acetylcholine in delay- and effort-based decision making in humans" for consideration as a Research Article by PLOS Biology.

Your manuscript has now been evaluated by the PLOS Biology editorial staff as well as by an academic editor with relevant expertise and I am writing to let you know that we would like to send your submission out for external peer review.

Once your full submission is complete, your paper will undergo a series of checks in preparation for peer review. After your manuscript has passed the checks it will be sent out for review. To provide the metadata for your submission, please Login to Editorial Manager (https://www.editorialmanager.com/pbiology) within two working days, i.e. by Dec 21 2023 11:59PM.

Kind regards,

Christian

Christian Schnell, PhD

Senior Editor

PLOS Biology

cschnell@plos.org

---

## [Decision Letter · Decision Letter 1]

8 Feb 2024

Dear Dr Erfanian Abdoust,

Thank you for your patience while your manuscript "Distinct roles of dopamine and acetylcholine in delay- and effort-based decision making in humans" was peer-reviewed at PLOS Biology. It has now been evaluated by the PLOS Biology editors, an Academic Editor with relevant expertise, and by three independent reviewers. 

In light of the reviews, which you will find at the end of this email, we would like to invite you to revise the work to thoroughly address the reviewers' reports.

As you will see below, the reviewers are overall supportive and think that the presented data are novel and potentially interesting. However, they raise a number of concerns that will need to be addressed by providing clarifications, more careful wording, and some additional details about the methods and analyses (in particular about the models).

Given the extent of revision needed, we cannot make a decision about publication until we have seen the revised manuscript and your response to the reviewers' comments. Your revised manuscript is likely to be sent for further evaluation by all or a subset of the reviewers.

**IMPORTANT - SUBMITTING YOUR REVISION**

*Re-submission Checklist*

*Published Peer Review*

*PLOS Data Policy*

*Blot and Gel Data Policy*

Sincerely,

Christian

Christian Schnell, PhD, 

Senior Editor

PLOS Biology

cschnell@plos.org

REVIEWS:

Reviewer #1: This manuscript asks the interesting question on the role of dopamine and acetylcholine in delay and effort-based decision making. While there is already several work on the role of dopamine in for example effort-based decision making, further examination of the role of dopamine in delay discounting and especially of acetylcholine in decision making more broadly seems relevant for a better understanding of the role of these neurotransmitters and their potential interplay. The authors report several analyses and control analyses using state-of-the-art analytical approaches and discuss their findings in relation to previous work as well as mentioning some limitations of their study. While well written in general, I believe the specific contribution of this study beyond previous work could be mapped out a bit more clearly in the introduction, while the authors should also be cautious and nuanced in their interpretation and generalisation of their findings. In addition, a few methodological details and some further potential limitations of their study due to the experimental design could be included. Specific comments and suggestions are detailed below.

- At several points in the manuscript, such as the abstract, discussion etc., the authors talk about a domain-general role of dopamine and a domain-specific role of acetylcholine. Given that only two types of costs (effort and delay) were tested here in only two different tasks, this conclusion might be a bit too broad. Furthermore, given the main effects and interaction effects found in decisions, reaction times and stochasticity, I believe the conclusion drawn - and the statement made - that the acetylcholine receptor antagonist produced opposite effects as the dopamine receptor antagonist goes a bit too far (Also the fact that no correlations were found in the across-task relationship analyses might indicate that the effects are not fully opposing). Instead of drawing such broad conclusions, it may be better to stick to the actual findings when summarizing and interpreting them (and only briefly speculate about domain-general versus domain-specific as well as opposing effects).

- Somewhat relatedly, in their final conclusion (page 19) the authors suggest a reciprocal relationship between cholinergic and dopaminergic activity. Given that the two neurotransmitters were manipulated in separate sessions, such a relationship seems speculative. Furthermore, the authors mention a domain-general role of dopamine in integrating proximity and effort costs. Again, since proximity and effort costs were manipulated in separate tasks, such an integration cannot certainly be concluded from this experiment, strictly speaking. I would prefer it if this was formulated more cautiously. 

- The introduction could be extended by further detailing previous results on the effects of dopamine on effort sensitivity and reward sensitivity and by further detailing previous findings on the role of dopamine in delay discounting. In addition, previous research on potential related roles of acetylcholine could be explicitly referred to in the introduction. Providing a bit more detail on previous related research may help the reader to better understand the contributions and hypothesis of this manuscript.

- Since participants were required to exert the chosen effort on every trial (albeit briefly), it seems plausible that fatigue could have developed in the effort discounting task (see e.g. Müller et al., 2021, Nature Communications; Matthews et al., 2023, Cognition) and might have affected the results and potentially contributed to the differences observed between tasks. If not modelled explicitly, this should at least be mentioned in the discussion.

- Relatedly, could the authors clarify why the order of the tasks was fixed, and discuss whether the observed effects could also be confounded by a potential order effect (potentially due to fatigue, or boredom, satiety, etc.)? 

- A few things regarding the design and methodology could be clarified and described in more detail. For example 

i) Could the authors clarify why a fixed alternative was used in the delay discounting task but not in the effort discounting task, which also makes the dependent variables in the regression analyses somewhat hard to compare between the tasks? 

ii) How much time did participants have to make their decision in each task? 

iii) Did every participant see the same trials in the same sequence? 

iv) Page 31, line 731: Why was here, in contrast to the other analyses, no Bayesian framework used? And why was timepoint T1 not analysed/reported here, as described in the main manuscript text?

- Page 12, line 259: «decreased delay sensitivity observed in the effort discounting task analysis» seems to be a typo.

- There are some more typos in the manuscript that the authors might want to correct.

Reviewer #2: The authors tested the effects of two drugs, one dopaminergic (haloperidol) and one cholinergic (biperiden) on choices made by healthy volunteers in two tasks, one testing effort discounting and the other delay discounting of monetary rewards. The main results are opposite effects on effort discounting, the willingness to exert effort being decreased by haloperidol and increased by biperiden, plus a single effect of haloperidol on delay discounting.

To my knowledge, this precise combination of 2 tasks by 2 drugs has never been reported. The opposite effects on effort discounting are novel and interesting. The manuscript is clearly written, the analyses are sophisticated. The authors should be commended for the impressive set of physiological and self-report data that were collected to document drug effects. 

Major remarks:

1) The comparison between the two tasks is problematic. I noted that they are analyzed separately, but there is still an implicit comparison in the conclusions, for instance when stating that biperiden affects effort but not delay discounting, or that dopamine is domain-general and acetylcholine domain-specific. There are several reasons why the tasks cannot be compared: one was always done after the other (hence at different latencies relative to the peak of drug efficiency), they employed different amounts of reward, one option was fixed in one task and not in the other, choices were hypothetical in one task and not in the other, etc. As is clearly apparent in Fig. 2, choice behavior was quite different: while choices were equally sensitive to effort and reward in the first task, they were almost insensitive to delay and dominated by reward in the second task. These design issues cannot be changed, but the comparison can be avoided, for instance by focusing on effort discounting, which yielded the more convincing results.

2) The parameters that capture drug effects in choice models seem arbitrary. As only discount factors and choice stochasticity can change with drug conditions, the model fit would miss an effect on reward sensitivity, which may be expected given the model-free results and the literature on dopamine. It would also miss a change in the global willingness to accepts costs, irrespective of reward, effort or delay levels. My suggestion is to include a weight on reward in the value function and an intercept (bias) in the choice function, with a potential modulation of these two parameters by drug conditions. To preserve recoverability, some parameters could be common to the two tasks (e.g., inverse temperature).

Other remarks:

3) One the main objectives announced in the introduction is to disambiguate the role of dopamine in delay-based decision-making. This cannot be done by using the same type of task as was used in previous studies. What the authors mean is that they intend to add another dataset to an already large literature on dopamine and delay discounting. Besides, the authors should refrain from concluding about dopamine (or acetylcholine) in general, since they used drugs that are specific to a subtype of dopamine (or acetylcholine) receptors.

4) The 6 possible effects (main effect, reward sensitivity and cost sensitivity for the 2 tasks) should be illustrated in Fig. 3, to avoid the impression that results are arbitrarily picked.

5) The possible motor effects of dopamine are not taken into account in the effort model, and neither is the potential impact on fatigue related to repeated squeezing. These potential effects can be tested on the parameters of force pulses and their variation across task trials.

6) Session effects and drug-by-session interactions should be included in linear mixed-effect models. There is usually meta-learning with these tasks, such that sensitivity to drug effects might differ between the first and second sessions in within-subject designs. 

7) Results on response times are not bringing any additional insight. They could easily be moved to supplementary material.

8) The interpretation of haloperidol effects in terms of proximity is purely tautological. Explaining that participants prefer delayed rewards because they are less sensitive to temporal proximity is just paraphrasing the result. Besides, it is incoherent to state that dopamine is domain-general because it has a role in processing effort in one case and proximity in the other. If the role of dopamine is different in effort and delay discounting, then it is not domain general.

Reviewer #3: In this manuscript the authors studied the effect of dopaminergic and muscarinic antagonists on effort discounting and time discounting using two separate cognitive tasks. They used a within person design in which each participant performed both tasks in three separate occasions, receiving one of the three possible treatments (placebo, haloperidol, or biperiden) on each occasion. The authors analyzed the behavioral data using a GLMM approach and a computational modelling approach. They report that haloperidol and biperiden had opposite effects on the effort discounting task, with haloperidol decreasing and biperiden increasing the willingness to make effort to gain rewards. For the delay discounting, they found that only haloperidol had an effect by decreasing delay discounting whereas biperiden had no effect.

There are not many studies in humans contrasting the effects of manipulations of the dopaminergic and acetylcholinergic systems within the same participants. In this sense, these are interesting data that increases our understanding on how dopamine and acetylcholine modulate decision-making processes. However, I believe that the data is less clarifying on the role of these neurotransmitters as the authors imply in t

---

## [Decision Letter · Decision Letter 2]

24 May 2024

Dear Dr Erfanian Abdoust,

Thank you for your patience while we considered your revised manuscript "Distinct roles of dopamine and acetylcholine in delay- and effort-based decision making in humans" for publication as a Research Article at PLOS Biology. This revised version of your manuscript has been evaluated by the PLOS Biology editors, the Academic Editor and the original reviewers.

Based on the reviews and on our Academic Editor's assessment of your revision, we are likely to accept this manuscript for publication, provided you satisfactorily address the following data and other policy-related requests.

* We would like to suggest a different title to improve readability: "Dopamine and acetylcholine have distinct roles in delay- and effort-based decision making in humans"

* Please add the links to the funding agencies in the Financial Disclosure statement in the manuscript details.

* All research involving human participants must have been approved by the authors' Institutional Review Board (IRB) or an equivalent committee, and must have been conducted according to the principles expressed in the Declaration of Helsinki. Please state in the manuscript that your study has been conducted according to the principles of the Helsinki Declaration.

* DATA POLICY:

Regardless of the method selected, please ensure that you provide the individual numerical values that underlie the summary data displayed in the following figure panels as they are essential for readers to assess your analysis and to reproduce it: 2A

We expect to receive your revised manuscript within two weeks. 

*Published Peer Review History*

*Press*

Sincerely,

Christian

Christian Schnell, PhD

Senior Editor

cschnell@plos.org

PLOS Biology

Reviewer remarks:

Reviewer #2: The authors have carefully addressed all my concerns and made substantial revisions to the manuscript. The results obtained in the effort discounting task are convincing and interesting. I still find weak the evidence obtained in the delay discounting task, and related conclusions about a dissociation between drugs in that task (because effects are similar and tiny), or between tasks for a given drug (because tasks are not comparable). However, with the appropriate cautionary notes, I guess the manuscript would be acceptable.

As requested by the Editor, I also checked the responses to Reviewer 3. There were mostly demands of clarifications, which I think were adequately provided, and suggestions for a model comparison testing the necessity of drug-sensitive parameters, which was implemented (with a conclusive outcome).

---

## [Editor Report · Decision Letter 3]

14 Jun 2024

Dear Dr Erfanian Abdoust,

Thank you for the submission of your revised Research Article "Dopamine and acetylcholine have distinct roles in delay- and effort-based decision making in humans" for publication in PLOS Biology. On behalf of my colleagues and the Academic Editor, Raphael Kaplan, I am pleased to say that we can in principle accept your manuscript for publication, provided you address any remaining formatting and reporting issues. These will be detailed in an email you should receive within 2-3 business days from our colleagues in the journal operations team; no action is required from you until then. Please note that we will not be able to formally accept your manuscript and schedule it for publication until you have completed any requested changes.

PRESS

Sincerely, 

Christian

Christian Schnell, PhD

Senior Editor

PLOS Biology

cschnell@plos.org